# Omnidirectional 3D Scene Reconstruction from Single Image

**Ren Yang**
Microsoft Research Asia
yangren@microsoft.com

**Jiahao Li**
Microsoft Research Asia
li.jiahao@microsoft.com

**Yan Lu**
Microsoft Research Asia
yanlu@microsoft.com

## Abstract

Reconstruction of 3D scenes from a single image is a crucial step towards enabling next-generation AI-powered immersive experiences. However, existing diffusion-based methods often struggle with reconstructing omnidirectional scenes due to geometric distortions and inconsistencies across the generated novel views, hindering accurate 3D recovery. To overcome this challenge, we propose Omni3D, an approach designed to enhance the geometric fidelity of diffusion-generated views for robust omnidirectional reconstruction. Our method leverages priors from pose estimation techniques, such as MASt3R, to iteratively refine both the generated novel views and their estimated camera poses. Specifically, we minimize the 3D reprojection errors between paired views to optimize the generated images, and simultaneously, correct the pose estimation based on the refined views. This synergistic optimization process yields geometrically consistent views and accurate poses, which are then used to build an explicit 3D Gaussian Splatting representation capable of omnidirectional rendering. Experimental results validate the effectiveness of Omni3D, demonstrating significantly advanced 3D reconstruction quality in the omnidirectional space, compared to previous state-of-the-art methods. Project page: https://omni3d-neurips.github.io.

## 1 Introduction

3D scene reconstruction from 2D image plays a significant role in the future development of computer vision technologies, paving the way for the next era of AI immersive media. However, this task is inherently ill-posed due to the significant geometric ambiguity and limited information available from a single input image. While recent progress, particularly with diffusion-based models [2, 11, 48, 34], has shown promise in object-level 3D reconstruction [22, 41, 4],and also facilitates scene-level Novel View Synthesis (NVS) [45, 10, 51] and scene reconstruction [20, 3] with 3D Gaussian Splatting (3DGS) [12]. However, the existing diffusion-based methods often encounter significant hurdles when applied to omnidirectional 3DGS reconstruction of real-world scenes. The main obstacle is that the produced novel views tend to suffer from geometric distortions and critical inconsistencies, especially when synthesizing views far from the original input perspective. Standard models may misinterpret context in omnidirectional images due to non-uniform structures and optical properties different from perspective images. These inaccuracies in geometry and appearance across generated views fundamentally hinder the ability to recover a coherent and accurate 3DGS representation of the full omnidirectional scene.

To address this challenge, we introduce Omni3D, a novel approach specifically designed to tackle the geometric and content consistency issues inherent in diffusion-based view generation for omnidirectional 3D scene reconstruction. Our core idea is to explicitly incorporate and refine geometric constraints throughout the view generation process. We achieve this by leveraging strong priors obtained from state-of-the-art pose estimation techniques, such as MASt3R [17], to iteratively im-

39th Conference on Neural Information Processing Systems (NeurIPS 2025).

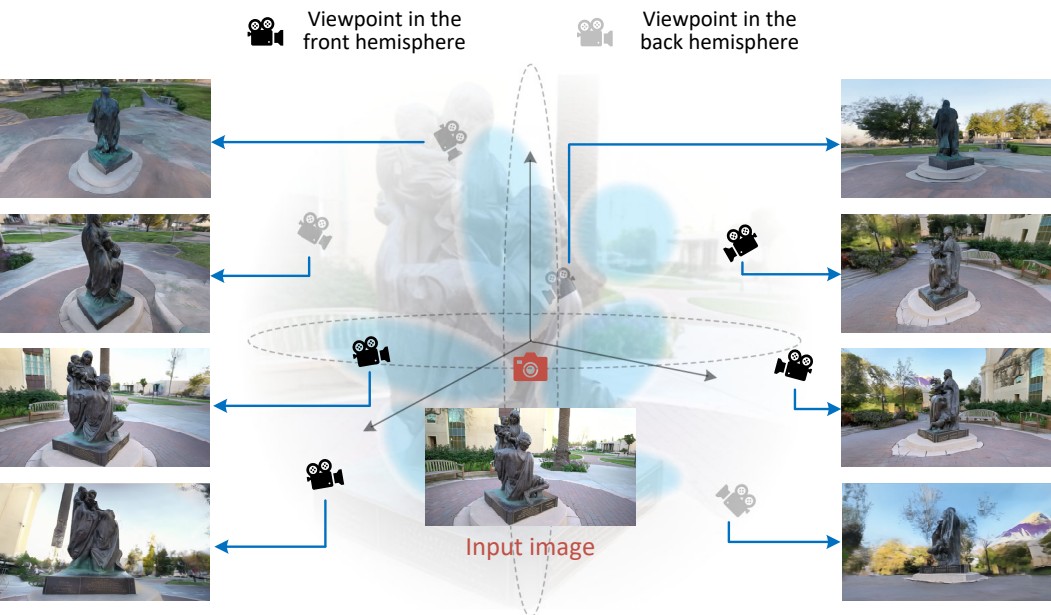

Figure 1: Example of Omni3D for omnidirectional 3D scene reconstruction from single image.

prove both the generated novel views and their corresponding camera pose estimations. Specifically, Omni3D employs a synergistic Pose-View Optimization (PVO) strategy. It minimizes 3D reprojection errors between pairs of generated views to progressively refine the image content, ensuring better geometric alignment. Simultaneously, the estimated camera poses for these views are corrected based on the feedback from the refined views. This iterative process of mutual refinement between view content and pose estimation yields a set of geometrically consistent novel views with highly accurate camera parameters. These consistent views and poses serve as high-quality input for building an explicit 3D scene representation using 3D Gaussian Splatting, enabling flexible rendering with advanced quality across the entire omnidirectional space. Figure 1 illustrates an example result, demonstrating omnidirectional 3D scene reconstruction from a single image by the proposed Omni3D approach.

Our primary contributions are concluded as:

- We propose a novel method, Omni3D, that significantly improves the geometric and content consistency of diffusion-generated novel views for single-image omnidirectional 3D scene reconstruction with Gaussian Splatting.

- We propose a synergistic Pose-View Optimization (PVO) process that leverages pose estimation priors to iteratively refine both generated view content and camera poses by minimizing 3D reprojection errors.

- We demonstrate state-of-the-art performance in omnidirectional 3D scene reconstruction from a single image, showing substantial improvements on rendering quality across a wide range of view angles compared to previous methods.

## 2 Related Works

### 2.1 Traditional View Synthesis

Early methods for Novel View Synthesis (NVS) often relied on geometric primitives or layered representations. Multiplane Images (MPI) [54, 36, 39, 47, 15, 49] represent a scene using multiple semi-transparent planes at different depths. Methods like SinMPI [26] and AdaMPI [8] extended MPIs for single-image NVS, sometimes incorporating diffusion models to hallucinate out-of-view content. However, MPI-based methods can struggle with representing complex non-planar geometry and may exhibit flatness artifacts. Another line of work [46, 31, 29, 35] utilizes depth-based warping,

where an estimated depth map is used to project the input view to a novel viewpoint, followed by inpainting occluded regions . These methods are highly sensitive to depth estimation errors and can produce artifacts near object boundaries or inconsistent content in inpainted areas. While effective for limited view changes, these traditional techniques often struggle with the large baselines and distortions inherent in omnidirectional reconstruction.

## 2.2 Generative Image-3D Reconstruction

Pre-trained text-to-image (T2I) diffusion models possess rich semantic and structural priors. Several works [30, 32, 27] aim to distill these priors for 3D generation and NVS. Some approaches [38, 25, 18, 42] optimize 3D representations using scores distilled from a 2D diffusion model as supervision. Others [21, 33] fine-tune 2D diffusion models to be conditioned on camera viewpoints, enabling them to generate novel views directly. While powerful, these methods often focus on object or simple scene reconstruction and may lack generalization to complex and large-scale scenes. Furthermore, controlling camera pose accurately can be challenging, as poses are treated as high-level prompts rather than precise geometric inputs. Besides, Chung *et al.* [5] and Yu *et al.* [50] employ diffusion-based inpainting models to lift 2D images to 3D scenes.

More recently, Latent Video Diffusion Models (LVDMs) [2] trained on large-scale video datasets have emerged as a promising source of 3D priors, implicitly learning about motion, temporal coherence, and scene dynamics. Several works [7, 14, 23, 24, 41] leverage LVDMs for single-image-to-3D reconstruction at the object level, and meanwhile, [45, 51] focus on scene-level 3D NVS and reconstruction. However, the inherent stochasticity and iterative denoising process in diffusion models may introduce geometric distortions and inconsistencies across generated views, hindering accurate 3D reconstruction in the scenario of large angle changes. The latest method LiftImage3D [3] employs distortion-aware Gaussian representations to mitigate view inconsistencies, but nevertheless it still reconstructs 3D in limited angles from the single input image, instead of the challenging task of omnidirectional 3D scene reconstruction.

## 2.3 Pose Estimation

In early years, Ummenhofer *et al.* [40] and Zhou *et al.* [53] proposed estimating depthmaps and relative camera pose given the grountruth camera intrinsic parameters. Later, the DUSt3R [43] method was proposed in 2024, which represents a significant departure from traditional pipelines. It is designed to perform camera pose estimation from unconstrained image collections, requiring no prior knowledge of camera intrinsics. In the same time of pose etimation, the camera intrinsics can also be calculated. In the following, MASt3R [17] is built upon the backbone of DUSt3R with a focus on local feature matching for improving image matching accuracy.

# 3 Method

## 3.1 Overall Framework

The overall framework of the proposed Omni3D is illustrated in Figure 2-(a), outlining a multi-stage approach to achieve omnidirectional 3D reconstruction from a single image.

In Stage I, beginning with a single input image (shown as 📷), we employ a Multi-View Diffusion (MVD) model to synthesize an initial set of novel views (📷). These views are generated along four cardinal orbits (left, right, up, and down) to broadly cover the frontal hemisphere relative to the input. Subsequently, the proposed Pose-View Optimization (PVO) module is applied to the generated views. The PVO module collaboratively refines the estimated camera poses and corrects the generated view content, thereby mitigating geometric distortions and inconsistencies inherent in the initial MVD outputs. During the PVO process, the camera intrinsic parameters are also calculated [43].

Stage II focuses on expanding the view coverage laterally. Key views from the periphery of the frontal hemisphere generated in Stage I (*e.g.*, the leftmost and rightmost views, depicted as 📷) serve as new conditional inputs for the MVD model. This step synthesizes additional novel views (📷) that extend into the left and right hemispheres. These newly generated views then undergo the PVO optimization to ensure their geometric accuracy and consistency.

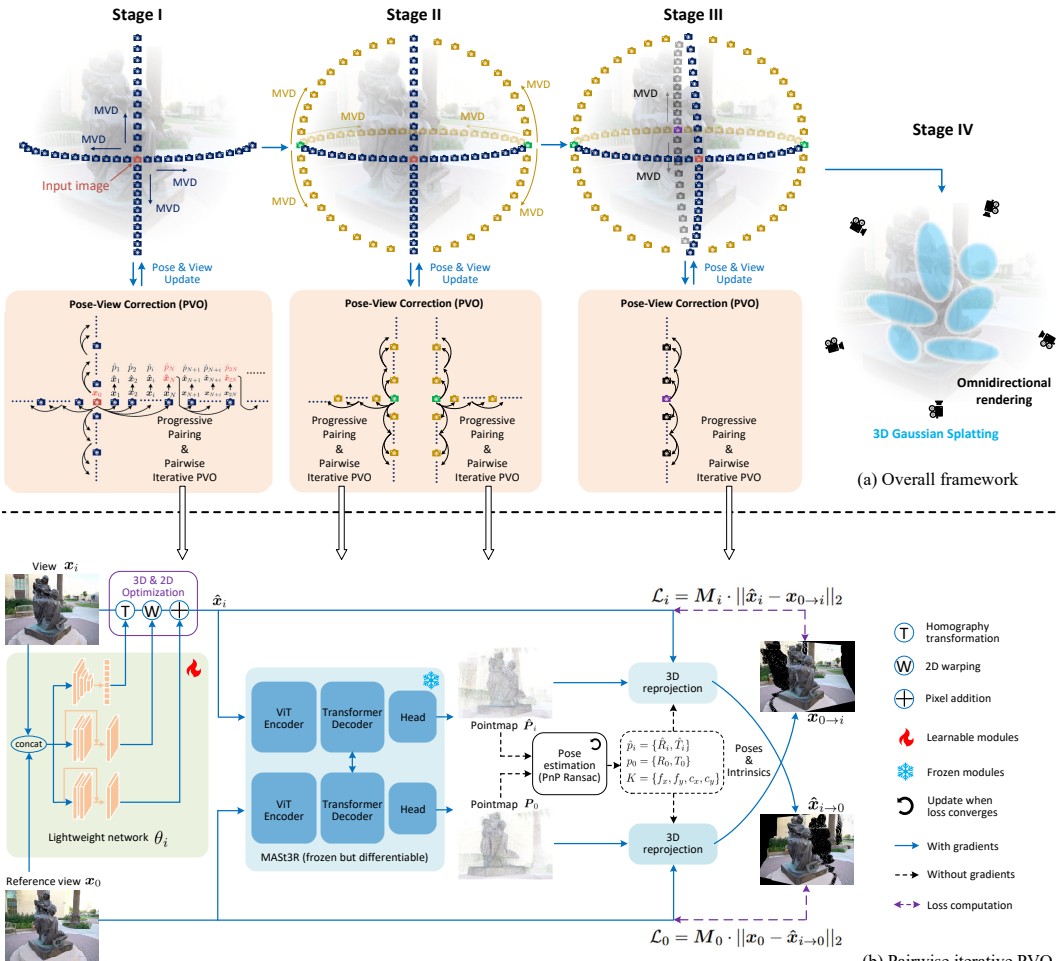

Figure 2: (a) The overall framework of the proposed Omni3D approach. Omni3D generates novel views in omnidirectional space across three stages. In each stage, the generated views are optimized by the Pose-View Optimization (PVO) module, which progressively applies a pairwise iterative PVO process to refine both view contents and pose estimation. This mitigates geometrical distortions and inconsistencies in the generated novel views, facilitating the final stage to represent the omnidirectional 3D scene with 3DGS. (b) The proposed pairwise iterative Pose-View Optimization (PVO) module. For each view pair, this module first estimates the initial camera poses and intrinsics, and then overfits a lightweight network for 3D and 2D optimization of a generated view by minimizing 3D reprojection errors between reference and generated views. Camera poses and camera intrinsics are then updated once the optimization loss converges. This process repeats iteratively for synergistically refining both the views and their corresponding poses.

Stage III addresses the back hemisphere to achieve fully omnidirectional coverage. In this stage, the backmost view (📷) is used to condition the MVD model for the synthesis of the final set of novel views (📷) required to complete the omnidirectional scene representation. As with the previous stages, these views are meticulously processed by the PVO module. Upon completion of this stage, a comprehensive set of geometrically consistent and pose-accurate omnidirectional views is obtained.

Finally, in Stage IV, this complete collection of PVO-optimized views, along with their refined camera poses and intrinsic parameters, is leveraged to reconstruct the 3D scene. Specifically, we train a 3D Gaussian Splatting (3DGS) model using these views. The resulting 3DGS model facilitates freely rendering of novel views in omnidirectional angles.

## 3.2 Multi-View Diffusion (MVD)

For the default implementation of Omni3D, we follow [37] to employ LoRA-tuned CogVideoX [48] as MVD models. These models are configured to generate 48 novel views per orbit, in addition to the original input view. The MVD models were trained on carefully selected samples from the DL3DV-10K dataset [19], with a strict separation maintained between the training and test sets. It is worth pointing out that Omni3D's effectiveness is not contingent on the choice of MVD model, and has the generalization capabilities across different MVD backbones, which is validated in Section 4.3.

## 3.3 Pose-View Optimization (PVO)

This section introduces our Pose-View Optimization (PVO) module, a core component of the Omni3D method. The PVO module employs a progressive pairing scheme to systematically process sequences of generated novel views. Coupled with this is an iterative optimization process designed to synergistically refine both the content of these views and their corresponding camera poses. This process yields views with enhanced geometric consistency and more accurate pose information, playing a critical role in facilitating robust omnidirectional 3D scene reconstruction with 3DGS.

### 3.3.1 Progressive Pairing

The progressive optimization process, illustrated in Figure 2-(a), is applied to each view generation orbit. To illustrate, we describe the procedure for a single orbit (e.g., the right orbit in Stage I), and views within all other orbits across Stages I, II, and III are processed analogously.

Let $x_0$ denote the initial input view for an orbit, and let $\{x_i\}_{i=1}^{I}$ represent the sequence of $I$ novel views generated along this orbit. The optimization proceeds in a sliding window manner. Initially, $x_0$ serves as the reference view and is paired with the first $N$ generated novel views, $\{x_i\}_{i=1}^{N}$. For each pair $(x_0, x_i)$ where $i \in \{1, \ldots, N\}$, the novel view $x_i$ undergoes our pairwise iterative Pose-View Optimization (PVO), as detailed in Section 3.3.2. This step yields an optimized view $\hat{x}_i$ and its corresponding refined pose $\hat{p}_i$. After this initial set of $N$ views is optimized, the $N$-th optimized view, $\hat{x}_N$ (along with its pose $\hat{p}_N$), becomes the new reference view. This new reference, $\hat{x}_N$, is then paired with the subsequent block of $N$ views, i.e., $(\hat{x}_N, x_{N+i})$ for $i \in \{1, \ldots, N\}$. These pairs then undergo the same PVO process. This progressive, sliding-window optimization scheme continues until all $I$ generated views within the orbit have been processed and refined.

This strategy is adopted to balance two competing factors. We note that consistently using the initial global input view as the reference for all pairs across an entire orbit would lead to progressively larger viewpoint disparities. Such large angular differences can significantly challenge the robustness of pose estimation and the efficacy of the PVO. Conversely, using each immediately preceding optimized view $\hat{x}_{i-1}$ as the reference for the current view $x_i$ might introduce error accumulation and propagation along the generation path of orbits.

In Omni3D, we address this trade-off by empirically setting the window size $N = I/4$. For our default setting where $I = 48$ views are generated per orbit, $N$ is therefore set to 12. This approach ensures that the maximum angular difference between the reference view and any target view within its optimization window remains manageable (*e.g.*, approximately $22.5°$ under our default settings), facilitating a stable PVO process.

### 3.3.2 Pairwise Iterative PVO

**Framework.** To simplify the notations, we use the pair $(x_0, x_i)$ as an example to introduce the pairwise iterative PVO network. The same process is utilized to all other pairs. As Figure 2-(b) illustrated, we overfit a lightweight $(\theta_i)$ network[1] on view pair $(x_0, x_i)$ to learn a Homography matrix ($\mathbf{H}$), a flow map ($\mathbf{F}$) and residual ($\mathbf{R}$) for 3D and 2D optimization ($\mathcal{O}$) of the generated view $x_i$. The optimized view $\hat{x}_i$ can be expressed as:

$$\hat{x}_i = \mathcal{O}(x_i, \theta_i) = \mathcal{W}(\mathcal{T}(x_i, \mathbf{H}), \mathbf{F}) + \mathbf{R}, \tag{1}$$

---

[1]As Figure 2-(b) shows, the Homography branch contains three CNN layers with strides of 1, 2 and 2, respectivaly, and a fully connected layer. The flow and residual branches have a residual block and an output layer in each. All CNN layers use the kernal size of 3 with 32 channels, except the output layers for flow and residual, whose channel numbers are 2 and 3, respectively.

where $\mathcal{T}(\cdot, \mathbf{H})$ and $\mathcal{W}(\cdot, \mathbf{F})$ denotes the Homography transformation and 2D warping, respectively. It is worth point out that each network ($\theta_i$) is overfit for each pair $(\boldsymbol{x}_0, \boldsymbol{x}_i)$ in an online training manner, and the weights do not share across pairs. Besides, it is important that the parameters in the lighweight network is zero-initialized, expect the bias of the output layer for Homography matrix, which initially outputs $\mathbf{I}_{3\times3}$. As such, the refined view $\hat{\boldsymbol{x}}_i$ is initialized as the input $\boldsymbol{x}_i$, *i.e.*,

$$\hat{\boldsymbol{x}}_i^{\text{init}} = \boldsymbol{x}_i, \quad \text{given} \quad \mathbf{H} = \mathbf{I}_{3\times3}, \mathbf{F} = \mathbf{0}, \text{ and } \mathbf{R} = \mathbf{0}. \tag{2}$$

Next, we employ the MASt3R [17] network to produce the pointmaps of $\boldsymbol{x}_0$ and $\hat{\boldsymbol{x}}_i$ (initially $\boldsymbol{x}_i$) as $\boldsymbol{P}_0$ and $\hat{\boldsymbol{P}}_i$, respectively. Note that $\boldsymbol{P}_0$ and $\hat{\boldsymbol{P}}_i$ are presented in a some coordinate, which we denote as the world coordinate. Then, the Perspective-n-Point (PnP) [9, 16] pose computation method with RANSAC [6] scheme is applied to estimate the camera poses (camera-to-world), which are defined as $p_0$ and $\hat{p}_i$ for the views $\boldsymbol{x}_0$ and $\hat{\boldsymbol{x}}_i$, respectively. Meanwhile, we can also obtain the camera intrinsics $K$ given the estimated poses [43], containing focal values ($f_x$, $f_y$) and the coordinates of the principal point ($c_x, c_y$).

In the following, with the input view $\boldsymbol{x}_0$, its pointmap $\boldsymbol{P}_0$, the pose of the target view $\hat{p}_i$ and camera intrinsics, we are able to reproject $\boldsymbol{x}_0$ to the target view in the 3D space. Specifically, we first convert the camera pose $\hat{p}_i$ to the world-to-camera matrix $\hat{p}_i'$, *i.e.*,

$$\hat{p}_i' = \begin{pmatrix} \hat{R}_i' & \hat{T}_i' \\ \mathbf{0}^T & 1 \end{pmatrix} = \hat{p}_i^{-1} = \begin{pmatrix} \hat{R}_i & \hat{T}_i \\ \mathbf{0}^T & 1 \end{pmatrix}^{-1}, \tag{3}$$

and then transform the pointmap $\boldsymbol{P}_0$ to the target view's coordinate

$$\boldsymbol{P}_0' = \hat{R}_i' \boldsymbol{P}_0 + \hat{T}_i', \tag{4}$$

to reproject it into the 2D screen coordinate $(u_i, v_i)$ of the target view using camera intrinsics, *i.e.*,

$$\begin{pmatrix} \tilde{u}_i \\ \tilde{v}_i \\ z_i \end{pmatrix} = K\boldsymbol{P}_0' = \begin{pmatrix} f_x & 0 & c_x \\ 0 & f_y & c_y \\ 0 & 0 & 1 \end{pmatrix} \begin{pmatrix} X \\ Y \\ Z \end{pmatrix}, \quad \text{where } \boldsymbol{P}_0' = \begin{pmatrix} X \\ Y \\ Z \end{pmatrix}, \tag{5}$$

and we define $(u_i, v_i) = (\tilde{u}_i/z_i, \tilde{v}_i/z_i)$. Finally, the RGB values $\boldsymbol{x}_0$ of each 3D point of $\boldsymbol{P}_0$ can be mapped into its projected location $(u_i, v_i)$ in the target view's coordinate, considering the depth $Z$ for visibility and blending overlapping points, *i.e.*,

$$\boldsymbol{x}_{0\to i} = \text{Render}\big((u_i, v_i), \boldsymbol{x}_0, Z\big). \tag{6}$$

Similarity, the 3D reprojection from $\hat{\boldsymbol{x}}_i$ to $\boldsymbol{x}_0$, *i.e.*, $\hat{\boldsymbol{x}}_{i\to0}$, can be calculated in the same manner[2].

After obtaining $\hat{\boldsymbol{x}}_{i\to0}$ and $\boldsymbol{x}_{0\to i}$, we define the loss function as

$$\mathcal{L} = \underbrace{\boldsymbol{M}_0 \cdot ||\boldsymbol{x}_0 - \hat{\boldsymbol{x}}_{i\to0}||_2}_{\mathcal{L}_0} + \underbrace{\boldsymbol{M}_i \cdot ||\hat{\boldsymbol{x}}_i - \boldsymbol{x}_{0\to i}||_2}_{\mathcal{L}_i}, \tag{7}$$

where $\boldsymbol{M}_0$ and $\boldsymbol{M}_i$ mask out the black pixels resulting from occlusion. The loss function is minimized to overfit the lightweight network for refining the generated view in an online training manner. Note that, during the training process, the MASt3R network itself remains unchanged, but its differentiability is crucial as it allows error back-propagation.

**Iterative optimization.** In the proposed PVO method, we employ an iterative optimization scheme to jointly optimize the generated view and refine the estimated camera pose. Recall that at the beginning of optmization, we have $\hat{\boldsymbol{x}}_i^{\text{init}} = \boldsymbol{x}_i$ and we calculate the initial pose estimations ($p_0$ and $\hat{p}_i$) and camera intrinsics. Given these parameters, we train the lightweight network $\theta_i$ to optimize the generated view $\hat{\boldsymbol{x}}_i$ by minimizing the loss function defined in Equation (7). During this training phase, the estimated poses ($p_0$ and $\hat{p}_i$) and camera intrinsics are held constant, and only $\hat{\boldsymbol{x}}_i$ is optimized. Once convergence is observed for the view optimization, we update the poses $p_0$ and $\hat{p}_i$ based on the refined view, and also camera intrinsics. This updated pose and intrinsics are then used in the next iteration to further optimize $\hat{\boldsymbol{x}}_i$. This cycle of optimizing the view with fixed poses, followed

---

[2]For the 3D reprojection operations, we utilize PyTorch3D [28], which provides differentiable rasterization and rendering functions.

by updating the poses and intrinsics based on the refined view, is repeated until the estimated poses converge.

This iterative refinement process simultaneously addresses geometric distortions and inconsistent content in $\hat{x}_i$ and improves the accuracy of the pose estimation. In our experiments, we empirically observed that the estimated poses consistently converge after three updates (in addition to the initial pose estimation). Consequently, we set the number of iterations to 3 in our proposed PVO method.

### 3.3.3 Parallelism

Leveraging the pairing scheme introduced in Section 3.3.1, the PVO process of view pairs that share the same reference view are independent from each other, allowing significant parallelism in computation. To be specific, in Stage I, the PVO of at most $4N$ pairs can be computed in parallel. In Stages II and III, there are at most $3N$ and $2N$ pairs can be optimized concurrently, respectively. In our experiments, the machine with 8 NVIDIA A100 GPUs allows the paralleled PVO of 24 pairs. Hence, given $N = 12$, we are able to parallelly compute PVO on $N$ pairs of two orbits. This way, the entire framework requires only 24 serial PVO computations across all stages, *i.e.*, 8, 12 and 4 serial PVOs in Stages I, II and III, respectively. This does not significantly increase the overall computational time. We analyze the time consumptions in Section 4.5.

## 4 Experiments

### 4.1 Evaluation Protocol

The proposed Omni3D method represents the reconstructed 3D scene using 3D Gaussian Splatting (3DGS). Consequently, we evaluate the reconstruction performance by rendering views from the 3DGS model at the camera poses corresponding to the groundtruth views and comparing these rendered images to the groundtruth. To facilitate this comparison, it is essential to align the 3D coordinates of groundtruth with the MASt3R coordinates employed in our method. Therefore, we associate each groundtruth view with four specific reference views used in Omni3D, depicted as 📷, 📷 and 📷 in Figure 2-(a), and then utilize MASt3R to estimate the pose of each groundtruth view. During this pose estimation process, the poses of the four selected Omni3D reference views are held fixed. By doing so, we effectively align the estimated poses of the groundtruth views to our established MASt3R coordinate system. This alignment allows us to render images from 3DGS at the poses of the groundtruth for evaluation. It is crucial that after aligning the coordinates of groundtruth, the groundtruth views are not included for the training of 3DGS, and are only used in evaluation.

### 4.2 Experimental Setup

We quantitatively evaluate the 3D scene reconstruction quality of Omni3D on three distinct datasets: the Tanks and Temples [13], Mip-NeRF 360 [1], and DL3DV [19] datasets. For Tanks and Temples [13] and Mip-NeRF 360 [1], we evaluate Omni3D on their whole test sets. For DL3DV [19], we randomly select test scenes non-overlapping with the training samples of MVD. In each test sample, we randomly select groundtruth views in the entire omnidirectional space. We compare the performance of the proposed Omni3D against previous state-of-the-art open-sourced methods, including ZeroNVS [33], ViewCrafter [51], and LiftImage3D [3], in terms of PSNR, SSIM [44] and LPIPS [52]. Furthermore, we conduct comprehensive ablation studies to validate the effectiveness of the proposed PVO and analyze the computational time required for each component of the Omni3D framework. More experimental results are in **Appendix**.

### 4.3 Main Results

Table 1 presents a quantitative summary of our results. As demonstrated in the table, our proposed Omni3D approach consistently outperforms all compared methods across all evaluated datasets and metrics. Notably, on the Tanks and Temples dataset, we achieve a significant PSNR improvement of 1.45 dB compared to the recent LiftImage3D [3], and approximately 2.4 dB over ViewCrafter [51]. For perceptual quality metrics SSIM and LPIPS, Omni3D also clearly surpasses the compared methods. Similar performance gains are also observed on the Mip-NeRF 360 and DL3DV datasets, where Omni3D advances the PSNR results of LiftImage3D by 1.62 dB and 0.87 dB, respectively, and

Table 1: Evaluation of rendered views in omnidirectional space.

| Methods | Tanks and Temples | | | Mip-NeRF 360 | | | DL3DV | | |
|---|---|---|---|---|---|---|---|---|---|
| | PSNR ↑ | SSIM ↑ | LPIPS ↓ | PSNR ↑ | SSIM ↑ | LPIPS ↓ | PSNR ↑ | SSIM ↑ | LPIPS ↓ |
| ZeroNVS [33] | 12.67 | 0.4647 | 0.7506 | 13.40 | 0.2413 | 0.8299 | 11.28 | 0.4725 | 0.7074 |
| ViewCrafter [51] | 13.91 | 0.4714 | 0.5886 | 14.06 | 0.2420 | 0.7649 | 16.61 | 0.6185 | 0.3883 |
| LiftImage3D [3] | 14.85 | 0.4841 | 0.5781 | 14.27 | 0.2491 | 0.6479 | 16.21 | 0.6020 | 0.4844 |
| **Our Omni3D** | **16.30** | **0.5308** | **0.5166** | **15.89** | **0.2859** | **0.6369** | **17.08** | **0.6649** | **0.3348** |

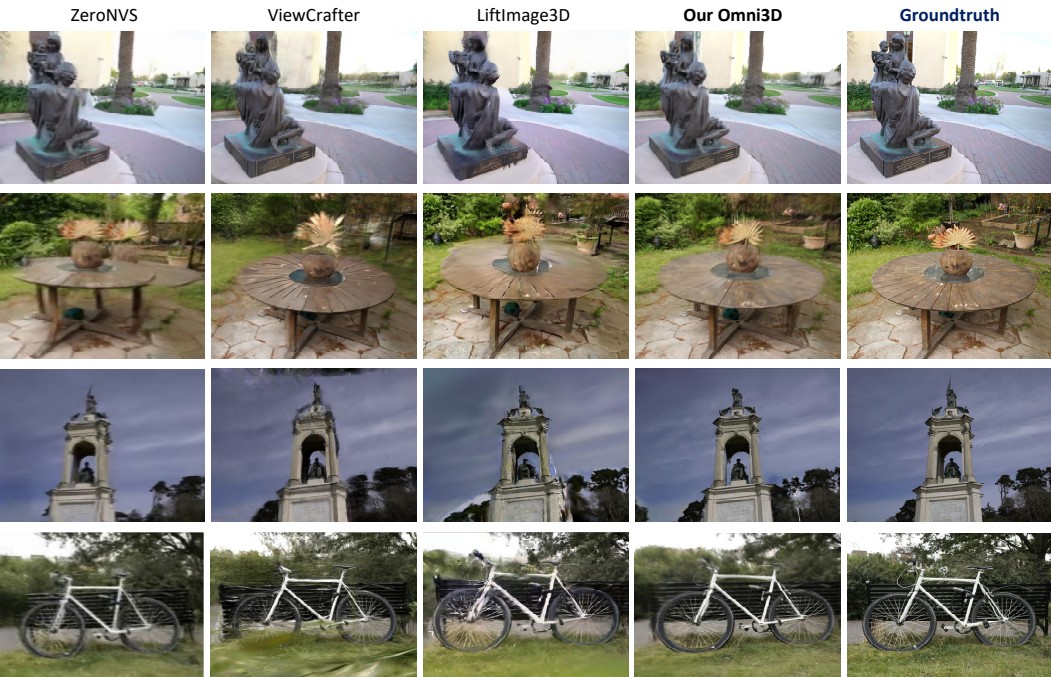

Figure 3: Visual results of rendered views of our Omni3D and compared approaches.

Table 2: Results of user study on our Omni3D and compared methods.

| Methods | Tanks and Temples | Mip-NeRF 360 | DL3DV |
|---|---|---|---|
| ZeroNVS [33] | 1.0 | 1.3 | 0.8 |
| ViewCrafter [51] | 4.3 | 4.7 | 7.4 |
| LiftImage3D [3] | 5.1 | 4.5 | 5.8 |
| **Our Omni3D** | **7.6** | **7.9** | **8.2** |

also achieve superior perceptual quality. We illustrate the visual results in Figure 3. In comparison with pervious methods, our Omni3D approach achieves rendered views with higher visual quality, less distortions or artifacts, and better geometrical accuracy to the groundtruth. These visual results align with the conclusion from the numerical evaluations, validating the effectiveness of Omni3D.

We further conducted a user study with 10 non-expert users, who are requested to rate the reconstructed 3D scenes with scores from 0 (poorest quality) to 10 (perfect quality). In the user study, we render the images with omnidirectional trajectories from the 3DGS generated by our Omni3D and the compared methods and send them to the users in video format, to reduce the hardware requirements for the users' personal computers. The average ratings are shown in Table 2. It can be seen from Table 2 that our Omni3D approach has obvious superior perceptual quality performance, which aligns with the numerical results.

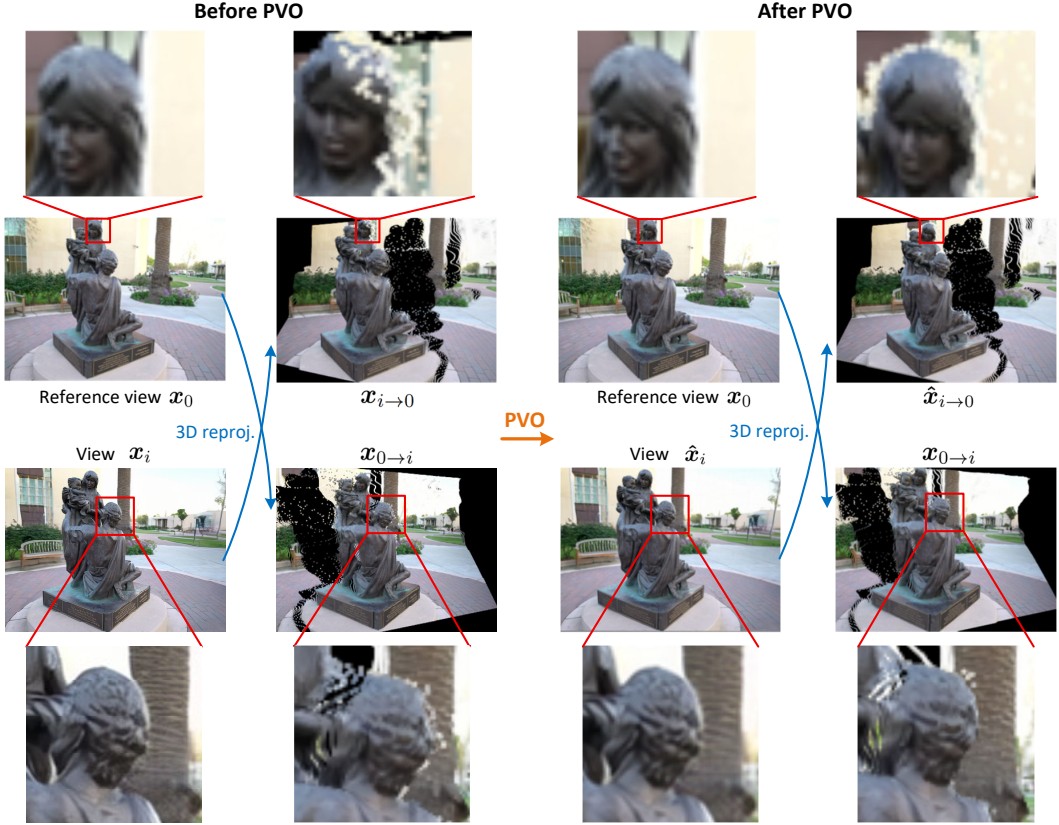

Figure 4: Ablation visual example of 3D-reprojected views before (left) and after (right) PVO.

Table 3: Ablation study on MVD and PVO

|  | PSNR ↑ | SSIM ↑ | LPIPS ↓ |
|---|---|---|---|
| Omni3D w/o PVO | 15.56 | 0.5198 | 0.5346 |
| **Omni3D** | **16.30** | **0.5308** | **0.5166** |
| LiftImage3D [3] | 14.85 | 0.4841 | 0.5781 |
| **LiftImage3D + PVO** | **15.28** | **0.4964** | **0.5446** |

## 4.4 Ablation study

Figure 4 visually illustrates the effects of the proposed PVO method. It can be seen from Figure 4 that before PVO, the reprojected images $x_{i \to 0}$ and $x_{0 \to i}$ contain obviously difference on the objects' positions in comparison with the target views. For example, the relative position between the women's head and the background in $x_0$ and $x_{i \to 0}$, and the relative position between the man's head and the tree in $x_i$ and $x_{0 \to i}$ (highlighted by red squares). These indicate the geometrical inconsistency in these views. This inconsistency may be not very noticeable in NVS, however, may significantly challenge the accurate reconstruction with 3DGS. After applying the proposed PVO, it can be seen that the geometrical error in the 3D-reprojected views has been effectively corrected, reflecting the improved consistency between the optimized view $\hat{x}_i$ and the reference view $x_0$. As such, the PVO facilitates our Omni3D achieving state-of-the-art performance on omnidirectional 3D reconstruction.

Table 3 shows numerical results for the ablation studies on the Tanks and Temples dataset [13]. In comparison with the baseline model without PVO (Omni3D w/o PVO), the proposed Omni3D achieves $0.74$ dB improvement in terms of PSNR, and also better SSIM and LPIPS performance. Moreover, Table 3 also indicated the generalizability of the proposed PVO method to various MVDs. We conduct experiments for utilizing PVO on LiftImage3D [3], whose MVD employs MotionCtrl [45].

Table 4: Time consumption on $8\times$A100 GPUs.

|  | MVD | Pose calc. | 3DGS | Total |
|---|---|---|---|---|
| ZeroNVS [33] | - | - | - | 133.7 min |
| ViewCrafter [51] | 2.1 min | - | 12.8 min | 14.9 min |
| LiftImage3D [3] | 3.5 min | 1.5 min | 67.4 min | 72.4 min |
| **Our Omni3D** | 10.8 min | 10.5 min | 12.8 min | 34.1 min |

Table 5: Time consumption on a single A100 GPU.

|  | MVD | Pose calc. | 3DGS | Total |
|---|---|---|---|---|
| ZeroNVS [33] | - | - | - | 133.7 min |
| ViewCrafter [51] | 4.3 min | - | 12.8 min | 17.1 min |
| LiftImage3D [3] | 12.0 min | 1.5 min | 67.4 min | 80.9 min |
| **Our Omni3D** | 21.6 min | 83.9 min | 12.8 min | 118.3 min |

Table 3 shows that the proposed PVO method successfully advances the performance of LiftImage3D by 0.37 dB in terms of PSNR, and also benefits the SSIM and LPIPS performance.

### 4.5 Time consumption

Table 4 shows the analysis on the time consumption of each component of our Omni3D and compared methods. We conduct experiments on a machine with 8 NVIDIA A100 GPUs. In the compared methods, ZeroNVS [33] employs Score Distillation Sampling (SDS) for novel view generation. Since NeRF distillation is very time-consuming, it takes $> 2$ hours for the inference of ZeroNVS. In our Omni3D, we utilize a standard 3DGS, whose training is much faster than the distortion-aware 3DGS in LiftImage3D [3]. Most importantly, thanks to the parallelism scheme of Omni3D introduced in Section 3.3.3, the proposed PVO method does not significantly increase the total time consumption. The paralleled PVO takes 10.5 min, which is less than the time used for 3DGS optimization. As a result, the whole framework of Omni3D takes about 34 min for an omnidirectional 3DGS reconstruction from single image. It is obviously faster than ZeroNVS and LiftImage3D. Note that we also run the compared methods with maximal parallelism on the same hardware as ours.

In Table 5, We further show the results on a single A100 GPU. It can be seen from Table 3 that when the parallelism capability is limited to a single A100 GPU, the proposed Omni3D approach consumes additional 46.2% of computational time compared with LiftImage3D [3], and still faster than ZeroNVS [33] (CVPR'24). However, our Omni3D approach is able to reconstruct the entire omnidirectional 3D space, instead of only the forward-facing views, and also achieves better reconstruction quality.

## 5 Conclusion

In this paper, we introduced Omni3D, a novel framework specifically designed to enhance the geometric accuracy of diffusion-generated views for robust omnidirectional 3D Gaussian Splatting. Our key innovation lies in the synergistic Pose-View Optimization (PVO) process. By iteratively refining both the generated view content and their estimated camera poses, guided by geometric priors from techniques like MASt3R and minimizing 3D reprojection errors, Omni3D produces a set of geometrically consistent novel views with high pose accuracy. These refined views and poses serve as advanced foundation for constructing an explicit 3D Gaussian Splatting representation. As demonstrated by our experimental results, Omni3D achieves state-of-the-art performance in single-image omnidirectional 3D scene reconstruction, yielding substantially improved rendering quality across a wide range of view angles compared to existing methods. This work represents a significant step towards enabling accurate and high-quality 3D reconstruction of complex, omnidirectional environments from a single image.

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

# Appendix

## A Detailed Network Architecture

Figure 5 illustrates detailed architecture of the lightweight network in our PVO method. In this figure, the convolutional layers are denoted as "Conv, filter size, filter number", and we use $\downarrow 2$ to denote the stride of 2. "GeLU" indicates the activation function of Gaussian Error Linear Unit. Besides, the output layer in the Homography branch is a dense layer with 8 nodes, whose outputs are defined as $O_1 \sim O_8$, respectively. They form the Homography matrix $\mathbf{H}$ as

$$\mathbf{H} = \begin{pmatrix} O_1 & O_2 & O_3 \\ O_4 & O_5 & O_6 \\ O_7 & O_8 & 1 \end{pmatrix}. \tag{8}$$

Recall that all parameters in convolutional layers and weights in the dense layer are zero-initialized and the bias of the dense layer are initialized as $[1, 0, 0, 0, 1, 0, 0, 0]^T$. As such, we have

$$\hat{\boldsymbol{x}}_i^{\text{init}} = \boldsymbol{x}_i, \quad \text{given} \quad \mathbf{H} = \mathbf{I}_{3 \times 3}, \mathbf{F} = \mathbf{0}, \text{ and } \mathbf{R} = \mathbf{0} \tag{9}$$

to initialize the refined view $\hat{\boldsymbol{x}}_i$ with its original input $\boldsymbol{x}_i$ at the beginning of PVO.

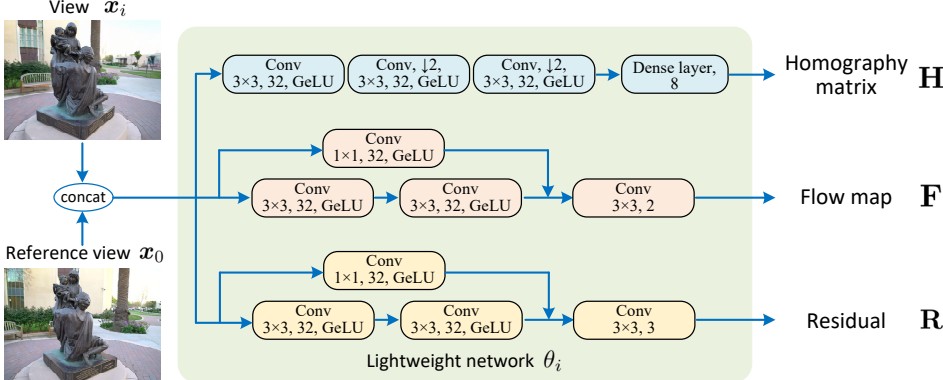

Figure 5: Detailed architecture of the lightweight network in PVO.

## B Additional Ablation Studies

**Ablation study on $N$ in progressive pairing.** Recall that in Section 3.3.1, we proposed a progressive pairing scheme with the window size of $N$, *i.e.*, in each generation orbit, the first $N$ generated views undergo PVO with the reference of $\boldsymbol{x}_0$, and then the next $N$ views takes the optimized $N$-th view $\hat{\boldsymbol{x}}_N$ as reference. This progressive scheme continues until all generated views within the orbit have been processed and refined. Table 6 shows the ablation study on the selection of $N$ on the Tanks and Temples dataset [13]. It can be seen from Table 6 that $N = 12$ results in the best performance. This outcome is likely attributable to the fact that utilizing views with large disparities as references (e.g., $N = 24$ or $48$) can significantly impair the robustness of pose estimation and diminish the efficacy of PVO. Conversely, employing each immediately preceding optimized view, $\hat{\boldsymbol{x}}_{i-1}$ as the reference for the current view $\boldsymbol{x}_i$ ($N = 1$), may inadvertently introduce error accumulation and propagation along the generation path of the orbits. Besides, setting $N = 1$ also considerably limits parallelism.

**Ablation study on iterations of pose updates in PVO.** Recall that in Section 3.3.2, we proposed a pairwise iterative PVO method. In the proposed PVO, given the initially estimated poses and intrinsics, we refine the generated view by minimizing the reprojection error until convergence, and then the poses and intrinsics are updated based on the refined views. This cycle is iteratively conducted until estimated poses converge. Table 7 illustrates the performance on the Tanks and Temples dataset [13] with different iterations of pose updates, in addition to the initial pose estimation. It can be seen that the performance converges at 3 iterations, *i.e.*, optimize the views and then update

Table 6: Ablation on progressive pairing

|  | PSNR ↑ | SSIM ↑ | LPIPS ↓ |
|---|---|---|---|
| w/o PVO | 15.56 | 0.5198 | 0.5346 |
| $N = 1$ | 16.24 | 0.5305 | 0.5170 |
| $N = 12$ | **16.30** | **0.5308** | **0.5166** |
| $N = 24$ | 16.19 | 0.5281 | 0.5179 |
| $N = 48$ | 15.98 | 0.5206 | 0.5254 |

Table 7: Ablation on iterations of pose updates in PVO, in addition to the initial pose estimation

| Iterations | PSNR ↑ | SSIM ↑ | LPIPS ↓ |
|---|---|---|---|
| 0 (w/o PVO) | 15.56 | 0.5198 | 0.5346 |
| 1 | 15.62 | 0.5207 | 0.5325 |
| 2 | 15.91 | 0.5254 | 0.5296 |
| 3 | 16.30 | 0.5308 | 0.5166 |
| 4 | 16.33 | 0.5311 | 0.5162 |

poses for three time. When the iteration number comes to 4, the performance negligibly increases. This verifies the reasonability for setting the iterations to 3 in our approach.

## C Limitations and Future Work

Current 3D reconstruction techniques, including the proposed Omni3D, ViewCrafter, and LiftImage, largely rely on a multi-stage process where 2D novel view synthesis acts as a crucial intermediary for generating 3D Gaussian Splatting (3DGS) reconstructions. This approach involves first synthesizing numerous 2D images from different perspectives and then using these synthesized views to infer the underlying 3D structure and appearance. While effective, this indirect methodology often introduces computational overhead and can sometimes limit the fidelity of the final 3D output due to potential errors or inconsistencies introduced during the 2D synthesis phase. The reliance on this intermediate step means that the overall efficiency and quality of 3D reconstruction are often constrained by the performance of the 2D view synthesis component.

The recent emergence of powerful foundation models across various domains presents a transformative opportunity for the field of 3D reconstruction. Instead of the current indirect approach, these advanced models could enable a more direct paradigm: training models to generate 3DGS, or even other sophisticated 3D formats, straight from a single 2D image input. This shift would fundamentally bypass the computationally expensive 2D intermediate steps, leading to several significant advantages. Not only could this potentially lead to a substantial improvement in the quality and realism of 3D reconstructions by eliminating potential bottlenecks introduced by the 2D synthesis, but it would also dramatically reduce the inference time. A direct generation pipeline would streamline the entire process, making 3D reconstruction faster and more accessible for a wider range of applications, from virtual reality to robotics.

Taking this vision even further, the ongoing development of world foundation models and AI agents hints at an even more revolutionary future. Imagine a scenario where complex 4D (3D over time) content could be generated directly from simple, high-level prompts, completely bypassing all 2D and 3D intermediaries. This would represent a paradigm shift, moving from data-driven reconstruction to concept-driven generation, where AI understands and creates dynamic 3D environments and objects based on abstract instructions. These ambitious directions, spanning from direct 3DGS generation to holistic 4D content creation, represent some of the most significant and exciting avenues for future research in artificial intelligence and computer graphics, promising to unlock unprecedented capabilities in digital content creation and spatial computing.

