# OpenReview forum: "Omnidirectional 3D Scene Reconstruction from Single Image"
_NeurIPS.cc/2025/Conference — NeurIPS 2025 poster_

### Official Review · Reviewer_xEyg · 2025-06-27

**Clarity:** 3
**Significance:** 2
**Originality:** 2
**Rating:** 4
**Confidence:** 4

**Summary:**

The paper introduces Omni3D, a framework for reconstructing a full 360° (omnidirectional) 3-D scene from a single RGB image.

A multi-view diffusion (MVD) model (here a LoRA-tuned CogVideoX) first hallucinates a dense set of novel views arranged on three progressively expanding hemispherical “orbits.”

A Pose–View Optimisation (PVO) module then iteratively refines each generated view and its camera pose simultaneously. PVO treats a pair of views, estimates an initial pose with MASt3R features, renders a 3-D reprojection between the pair, and trains a lightweight network to minimise reprojection error; the pose is re-estimated after each optimisation round.

After all views in all orbits are corrected, the set of images with accurate poses and intrinsics is used to train a standard 3D Gaussian splatting (3DGS) scene, enabling real-time omnidirectional rendering.

Experiments on Tanks & Temples, Mip-NeRF 360, and DL3DV-10K demonstrate that Omni3D outperforms ZeroNVS, ViewCrafter, and LiftImage3D, yielding better SSIM and LPIPS results.

**Questions:**

Please address the weakness listed above.

**Ethical Concerns:**

["NO or VERY MINOR ethics concerns only"]

**Final Justification:**

The authors’ substantial efforts in the rebuttal are commendable. Several of my concerns have been addressed—for instance, the evaluations on alternative camera distributions. The rebuttal also reinforces my initial impressions: the proposed pipeline appears overly complex, relies on unscalable per-pair “overfitted” refinement networks, and suffers from slow processing times (as shown in Table 3). Additionally, the requirement to train four separate MVDs further adds to the system’s complexity.

Overall, this work falls into the category of a system-level integration contribution. Although the methodological novelty may be limited, the development of this first framework capable of reconstructing a complete 360° 3D scene from a single RGB image can be appreciated. Thus, I upgrade my initial rating to a borderline accept.

**Quality:**

2

**Strengths And Weaknesses:**

Pros:
- It is new to demonstrably reconstruct an omnidirectional 3-D Gaussian scene from a single image. The synergistic PVO loop (jointly updating geometry and camera) is an interesting way to marry diffusion priors with classical reprojection error.

- Writing: The paper is easy to follow. Figures 2a/2b clearly step through the pipeline.

- Experimental: Three datasets, multiple SOTA baselines, three metrics, plus runtime analysis (Table 3). Ablations isolate PVO and show it generalises to a different backbone (MotionCtrl).

Cons:
- The entire pipeline is a bit cumbersome: MVD + per-pair refine network + MASt3R + PnP Ransac. Most of the components are prior art. The key component, PVO, is also a classical reprojection error-based optimization, although in the new context of point maps. Also, why do you design the refinement network according to the Homography transformation, which has a strong planar geometry assumption?

- The MVD part is only briefly mentioned. How is camera pose being controlled in the MVD, aligning with the target orbit? If the MVD generation somehow follows the orbit, why don't you make use of this initial camera pose in the subsequent geometry correction? Also, is the MVD module frozen?

- Per-pair “over-fitted” refinement network: The lightweight CNN used in PVO is independently over-fitted to every view pair. Because its parameters are never reused, it has zero cross-scene generalisation value. This raises the question: why not directly optimise geometric parameters (e.g., homographies, dense flow fields, or per-pixel depth offsets) and regularise them, instead of introducing an extra network?

Please clarify the motivation for choosing a network over a purely geometric parameterization. Add an ablation comparing (i) the current per-pair network, (ii) direct optimisation of a dense flow or homography with smoothness regularisation, and (iii) a shared network trained across multiple pairs;

- Evaluation restricted to omnidirectional targets: All quantitative results measure image quality at 360° target views. It remains unclear whether the proposed PVO+3DGS pipeline improves reconstruction quality when the target views lie in other camera distributions—e.g., the forward-facing used by ViewCrafter and LiftImage3D.

- The visual results in Fig. 3 still have many artifacts. This raises the question: Do we really need to make the task so hard? Some sparse-view based solutions like MVSplat360 can achieve much better 360 visual results, which seems more practical.

- Runtime fairness and resource usage: Table 3 reports wall-clock time using 8 × A100 GPUs, but it is not stated whether competing methods were benchmarked with identical hardware or scaled resources. As written, the comparison may conflate algorithmic efficiency with the advantage of more GPUs. Please provide runtimes for all methods on a single A100 (or specify each method’s GPU count and GPU hours) so that computational efficiency can be assessed fairly.

---

> ### Author Rebuttal · Authors · 2025-07-29
>
> We would like to extend our sincere thanks to the reviewer for the valuable and insightful comments, which are instrumental in improving our manuscript. Please find our point-by-point responses to each of the reviewers' comments in the following.
> ### **Overall pipeline and Homography**
> * **Overall pipeline.** We agree with the reviewer that the proposed pipeline has the potential to be simplified in future work. One possible way is generating 3D Gaussian splats straight from a single image input by a 3D fundamental model, bypassing the 2D (multi-view images) intermediate steps. This would not only simplify the overall pipeline but also significantly accelerate the inference speed.
> *  **Homography transformation.** Indeed, Homography transformation has a strong planar geometry assumption. However, it achieves image-level viewpoint transformation with only 8 parameters, and therefore can be used as an effective yet time-efficient way for the **coarse** correctness or alignment of camera views, especially when drifting viewpoint in a small angle. Beyond Homography, we also employ flow maps for pixel-level warping and learn residuals for the correctness of each pixel value, which play the role for **finer** correctness of generated views.  As such, we achieve a **hierarchical coarse-to-fine** process of pose-view optimization. Its effectiveness has been validated in the experiments.
> ### **Details about MVD**
> Thanks for the questions regarding MVD. We will describe the MVD more detailly in the revised manuscript.
> * **Training MVD for four orbits.** Instead of controlling MVD by camera poses, we have trained four MVD models for the four target orbits (left, right, up and down), respectively. Specifically, before training the MVD models, we first reconstruct the 3DGS of each sample in the training set, and then render it along the four target orbits to obtain the training samples of sequential images in the four orbits, respectively. Then, the MVD model for each orbit is trained by the samples of the corresponding orbit.
> * **Initial poses.** Although the MVDs show good performance for multi-view generation along the target orbits, it may inevitably occur slight errors between the generated orbits and the ideal orbits. Therefore, we employ MASt3R to calculate the initial poses of the generated views, instead of the poses predicted by the ideal orbits, to reduce the initial error of pose estimation before the PVO process.
> * **Frozen MVD.** After training the MVD models, they are frozen during the following processes, i.e., MASt3R, PVO and Gaussian optimization.
> ### **Design of the refinement network and ablation studies**
> Thanks a lot for the insights on the refinement network.
> * **Lightweight CNN vs. direct optimization of parameters.** In the proposed PVO process, we use a lightweight CNN to overfit the geometric parameters instead of directly optimize the parameters, since we observed instability (loss may increase) during the iterative optimization when directly optimize the geometric parameters themselves (denoted as w/o lightweight CNN). This is probably because that the lightweight CNN increases the nonlinearity of the network and are more compatible with the Adam optimizer, and therefore it is easier to adjust the learning rate during training. In the following, we show the ablation results in Table 1 in this rebuttal. It can be seen from Table 1 that directly optimizing the parameters (w/o lightweight CNN) leads to worse performance than the proposed lightweight CNN due to the worse convergence of PVO. Note that, in the case of instability, PVO has to be early stopped when the loss starts to increase, to avoid the crash of the whole pipeline.
>
>     \
>     Table 1: Ablation study on the lightweight CNN vs. direct optimization of parameters (w/o lightweight CNN)
>
>     | Methods | Tanks and Temples | Mip-NeRF 360 | DL3DV |
>     |:--------|:-----------------|:------------|:-----|
>     |         | PSNR $\uparrow\ \ \ $   SSIM $\uparrow\ \ \ $   LPIPS $\downarrow\ \ \ $ | PSNR $\uparrow\ \ \ $  SSIM $\uparrow\ \ \ $  LPIPS     $\downarrow\ \ \ $ | PSNR $\uparrow\ \ \ $  SSIM $\uparrow\ \ \ $  LPIPS $\downarrow$ |
>     | Omni3D (w/o lightweight CNN)|    15.92 $\ \ \ \ \ $ 0.5227 $\ \ \ \ $ 0.5175         |        15.21 $\ \ \ \ \ $ 0.2605 $\ \ \ \ $ 0.7262         |        16.57 $\ \ \ \ \ $ 0.6244 $\ \ \ \ $ 0.3890         |
>     | **Omni3D (w/ lightweight CNN)** |    **16.30** $\ \ \ \ \ $ **0.5308** $\ \ \ \ $ **0.5166**  |   **15.89** $\ \ \ \ \ $ **0.2859** $\ \ \ \ $ **0.6369**    |    **17.08** $\ \ \ \ \ $ **0.6649** $\ \ \ \ $ **0.3348**   |
>     | | | | | |
>
> * **Replace by a shared network.** If using a pre-trained (shared) network to refine the generated views according to estimated poses, it should learn to extract features from initially estimated poses and the corresponding initial re-projected images, in addition to the input image pair. The current lightweight CNN takes the inputs only the pair of reference and generated views without pose-related conditions, and therefore this network (or even its upscaled version) cannot play the role of a pre-trained network for the refinement of generated views. To employ a shared network, it is necessary to re-design the network structure. During the rebuttal period, it is hard to re-design and propose the pre-trained shared network, which are largely different from the proposed method. This may be an interesting topic that is valuable to be explored in future research.
>
> ### **Evaluation in other camera distributions**
> Thanks a lot for the valuable comments on the experimental setup. In this rebuttal, we have conducted additional experiments on the evaluation of rendering quality at forward-facing camera views, following the experimental setup of LiftImage3D. The results are shown in Table 2, where we can see that the proposed Omni3D approach also improves the reconstruction quality for the forward-facing camera views.
>
> \
> Table 2: Evaluation of rendered views at **_forward-facing camera distributions_**
>
> | Methods | Tanks and Temples | Mip-NeRF 360 | DL3DV |
> |:--------|:-----------------|:------------|:-----|
> |         | PSNR $\uparrow\ \ \ \ $   SSIM $\uparrow\ \ \ $   LPIPS $\downarrow\ \ \ $ | PSNR $\uparrow\ \ \ \ $  SSIM $\uparrow\ \ \ $  LPIPS     $\downarrow\ \ \ $ | PSNR $\uparrow\ \ \ \ $  SSIM $\uparrow\ \ \ $  LPIPS $\downarrow$ |
> | ZeroNVS [33]|         13.98 $\ \ \ \ \ $	0.4856 $\ \ \ \ $ 0.6225 |	14.25 $\ \ \ \ \ $	0.2669 $\ \ \ \ $	0.8657|	13.73 $\ \ \ \ \ $	0.5380 $\ \ \ \ $	0.6476
> | ViewCrafter [51]| 15.79 $\ \ \ \ \ $	0.5054 $\ \ \ \ $	0.5625	 | 15.99 $\ \ \ \ \ $	0.2717 $\ \ \ \ $	0.7012	 | 17.85 $\ \ \ \ \ $	0.6459	$\ \ \ \ $ 0.3843
> | LiftImage3D [3]|  16.14 $\ \ \ \ \ $	0.5232 $\ \ \ \ $	0.5346 | 	16.09 $\ \ \ \ \ $	0.2776 $\ \ \ \ $ 0.5553 |	20.77 $\ \ \ \ \ $	0.6834 $\ \ \ \ $	0.4557
> | **Our Omni3D** | **18.32** $\ \ \ \ \ $	**0.5919** $\ \ \ \ $ **0.4689** |	**16.96** $\ \ \ \ \ $	**0.2880** $\ \ \ \ $	**0.4807** | 	**22.02** $\ \ \ \ \ $	**0.7649** $\ \ \ \ $ **0.3011**
> | | | | | |
>
>
> ### **Task setting**
> Thanks a lot for the insightful comment on the task setting. Indeed, sparse-view based solutions have been widely studied in the community and have achieved promising results on 360-degree 3D reconstruction. Lifting a single image to the 3D representation (e.g., 3DGS) of the entire omnidirectional space is much more challenging, and meanwhile has the potential to be practical for the scenarios of AI-powered immersive media, such as the 3D experience of online meeting with mobile or laptop camera, and immersive 3D experience of the scene in a photograph. In recent years, there are a number of works in this direction, however, the existing works, e.g., LiftImage3D, ViewCrafter, etc., only effectively reconstruct 3DGS in limited angles, encountering significant hurdles to reconstruct omnidirectional 3D scenes. In this work, we pioneered in omnidirectional 3D reconstruction from single image, thanks to the proposed pipeline and the PVO method in our Omni3D approach.
> ### **Comparison of time consumption**
> Thank the reviewer very much for the valuable comments on the comparison of time consumption.
> * **Fairness for the comparison on 8 x A100 GPUs.** In the original manuscript, the proposed Omni3D and all compared methods are tested on **identical hardware**, i.e., 8 x A100 GPUs. Moreover, we have also applied **parallelism computation to all compared methods** as significant as possible, such as paralleled MVDs for various orbits, to fully leverage the capabilities of the hardware. Therefore, it ensures fairness in the comparison of time consumption. We will declare this issue more clearly in the revised manuscript.
> * **Time consumption on 1 x A100 GPUs.** Indeed, as suggested by the reviewer, it is necessary to report the computational time on a single GPU as well. We show the results in Table 3 in this rebuttal as below. It can be seen from Table 3 that when the parallelism capability is limited to a single A100 GPU, the proposed Omni3D approach consumes additional 46.2% of computational time compared with LiftImage3D, and still faster than ZeroNVS (_CVPR'24_). However, our Omni3D approach is able to reconstruct the entire omnidirectional 3D space, instead of only the forward-facing views, and also achieves better reconstruction quality. In the revised manuscript, we will include the time consumption results on a single GPU in the paper.
>
>     \
>     Table 3: Time consumption on a single A100 GPU
>     |                | MVD | Pose calc. | 3DGS | Total |
>     |----------------|-----|------------|------|-------|
>     | ZeroNVS [33]| -   | -          | -    |    133.7 min |
>     | ViewCrafter [51] |   4.3 min  | -          |   12.8 min   |    17.1 min   |
>     | LiftImage3D [3] |   12.0 min |      1.5 min      |  67.4 min    |   80.9 min   |
>     | **Our Omni3D** |  21.6 min   |      83.9 min      |   12.8 min   |   118.3 min    |
>     ||||

---

> > ### Comment · Reviewer_xEyg · 2025-08-06
> > **Partially address my concerns**
> >
> > The authors’ substantial efforts in the rebuttal are commendable. Several of my concerns have been addressed—for instance, the evaluations on alternative camera distributions. The rebuttal also reinforces my initial impressions: the proposed pipeline appears overly complex, relies on unscalable per-pair “overfitted” refinement networks, and suffers from slow processing times (as shown in Table 3). Additionally, the requirement to train four separate MVDs further adds to the system’s complexity.
> >
> > Overall, this work falls into the category of a system-level integration contribution. The methodological novelty may be limited. On the other hand, the development of such a framework capable of reconstructing a complete 360° 3D scene from a single RGB image can be appreciated.

---

> > > ### Author Response · Authors · 2025-08-06
> > >
> > > We would like to express our sincere gratitude to the reviewer. The reviewer's valuable and constructive insights are significantly beneficial and instrumental for strengthening the quality of our paper and providing promising direction for future research.

---

> > > ### Author Response · Authors · 2025-08-06
> > > **Further clarifications regarding complexity and novelty**
> > >
> > > Thanks again for the reviewer's follow-up comments on our rebuttal. Regarding the key concerns of the reviewer, we would like to make further clarifications regarding complexity and novelty.
> > >
> > > * **Complexity.** Indeed, the overall complexity of our framework is higher than the existing SOTA method, i.e., LiftImage3D. However, complexity is reasonable be comprehensively considered with the challenge of our task and reconstructible space compared with previous methods. We employ a more complex pipeline with relatively slower processing speed, since we aim at a much more challenging task and reconstruct the entire omnidirectional 3D space from a single image. On the contrary, LiftImage3D only reconstructs the forward-facing views in limited angles (shown as Figure 3 in the LiftImage3D paper). Although we consume 46.2% more computational time than LiftImage3D, the effective area and space that we can reconstruct is several times larger. Therefore, our **time efficiency per reconstructible space** is obviously higher than LiftImage3D. Besides, we also achieve faster speed than ZeroNVS (CVPR’24), whose reconstruction quality is much lower than ours.
> > >
> > > * **Novelty.** Our approach is not a simple integration of existing technologies. The core contribution of our paper is proposing a novel Pose-View Optimization (PVO) method which empowers multi-view generative model with the ability to reconstruct the omnidirectional space of 3D scenes with satisfied quality. In PVO, we novelly propose synergistically optimizing the generated contents and its estimated poses. Besides, the proposed lightweight network in PVO achieves hierarchical coarse-to-fine correctness of generated views and ensures effective and time-efficient convergence, and also facilitates parallelism. The proposed PVO method enables us to pioneer in effective omnidirectional 3D scene reconstruction from a single image.
> > >
> > >     Without the proposed PVO, the existing methods, such as LiftImage3D, are not able to be straightforwardly extended to omnidirectional 3D reconstruction, because LiftImage3D does not have the ability to mitigate errors during multi-view generation. Therefore, the accumulation of content distortions and geometrical inconsistencies prevents LiftImage3D from further extending the generation process to panorama with satisfied quality.
> > >
> > > We would like to thank the reviewer again, and appreciate any follow-up comments and discussions from the reviewer.

---

### Official Review · Reviewer_yK7z · 2025-06-30

**Clarity:** 3
**Significance:** 3
**Originality:** 3
**Rating:** 5
**Confidence:** 4

**Summary:**

This paper introduces a method to tackle the challenge of single-image 3D reconstruction by integrating the diffusion-based novel view synthesis methods and a pose refinement loop, named Pose-View Optimization (PVO). The proposed method progressively synthesizes sets of novel views around the input view, or around optimized views with a larger disparity to the input one. Then, the PVO module iteratively refines the pose pair of the novel views by PnP, where the 3D points are estimated from a pretrained MASt3R model. This core component yields a set of more geometry-consistent posed novel views. Finally, these pose-optimized views are fed into 3DGS pipelines to generate an omnidirectional scene representation.

**Questions:**

- Can this method generalize to scenes other than object-centric, e.g. forward-facing scene generation?
- How long does it take to optimize the lightweight network?
- How much does the optimized poses drift from the input ones? If not, why not register the four optimized poses (front, left, right, back) to the ground truth poses followed by evaluating the rendered views at registered poses?

**Ethical Concerns:**

["NO or VERY MINOR ethics concerns only"]

**Final Justification:**

This work presents a systematic way to reconstruct a 3D scene from a single image. The iterative PVO module, being the core module of this work, allows this method to generalize to different scenes, showing its potentially broad impact.

**Limitations:**

Yes

**Quality:**

4

**Strengths And Weaknesses:**

### Strengths
- An innovative design to overcome the geometry-conflicting distortion introduced by diffusion methods. The lightweight network within this design estimates the residual on top of the diffused view, which is further improved by jointly optimizing its pose.
- This method does not reply on the choice of the novel view diffusion methods, thereby providing a general framework that can be used for omnidirectional reconstruction from single view inputs.
- The splitting strategy allows a good parallelism and makes this method applicable.
- The manuscript is well written.
### Weakness:
- Highly dependent on the quality of MASt3R. If the point map estimated by the feed-forward method has poor quality, the whole pipeline may fail.
- L173, "expect" -> "except"

---

> ### Author Rebuttal · Authors · 2025-07-26
>
> We would like to thank the reviewer very much for the positive comments, and thanks a lot for the valuable questions and for pointing out the typo. We response to the reviewer's questions point-by-point as follows:
> ### **Generalizability on various kinds of scenes**
> In our experiments, we evaluated the proposed method on three commonly used datasets including various kinds of natural scenes, which are *not* limited to object-centric scenes. The generalizability of scene generation depends on training dataset of the Multi-View Diffusion (MVD). As described in Section 3.2, the MVD in our method is trained on DL3DV-10K, which is the largest scale dataset of 3D scenes as far as we know at the time point of submitting this manuscript. Employing larger-scale datasets, such as SceneSplat-49K (published in June 2025), may further enhance the generalizability of our approach. Besides, please note that the proposed Pose-View Optimization (PVO) method, which is the core contribution of this manuscript, is independent from the specific MVD, and can be generalized to various kinds of MVDs (as validated in Section 4.4).
> ### **Time for optimizing the lightweight network**
> Table 3 in our manuscript shows the analysis on the time consumption of each component of our Omni3D approach. The lightweight network is optimized in an online training manner during the proposed PVO scheme in our approach, which averagely takes 10.5 min on a test sample.  This time consumption is less than the time used for 3DGS optimization, and therefore, does not significantly increase the overall time consumption.
> ### **Evaluation protocol**
> Thanks a lot for the suggestion regarding the evaluation protocol. During the proposed PVO process, the poses of generated views only slight drift in most cases, probably because of the relatively high capacity of the Multi-View Diffusion (MVD) model.
> In theory, our evaluation protocol, that incorporate the groundtruth views into the coordinate of the generated and optimized views to align the coordinates, should be _equivalent_ to registering the optimized poses of generated views to the coordinate of groundtruth poses. However, our evaluation protocol has a benefit that it can be applicable to the evaluation with *unposed* groundtruth views.
> ### **Dependency on MASt3R**
> We agree to the reviewer that the proposed pipeline highly depends on a reliable pose estimation method. We utilize MASt3R in this manuscript, which performs effectively on all test datasets. Moreover, some advanced methods, e.g., MV-MASt3R, VGGT, etc, were proposed recently and have shown better performance than MASt3R, and therefore, they may potentially provide more reliable pose estimation to our proposed Omni3D approach.
> ### **Typos**
> Thanks a lot for pointing out the typo. We will undertake a thorough review of the entire manuscript to correct this and any other typographical errors before final submission.

---

> > ### Comment · Reviewer_yK7z · 2025-08-06
> >
> > Thank the authors for their comments. It seems that adding more details on generalizing to forward-facing scenes would strengthen the proposed work. Reviewer xEyg also raised similar questions. While the rebuttal presented the results on forward-facing scenes, it would be better to explain how to generate camera trajectories, which will be different from those introduced in the paper.

---

> > > ### Author Response · Authors · 2025-08-07
> > > **Generation trajectories for forward-facing scenes**
> > >
> > > We would like to thank the reviewer very much for the follow-up comments, and thanks for the valuable insights in adding the details for the forward-facing scenarios to strengthen our manuscript.
> > >
> > > In the case of forward-facing scenes, we use the multi-view generation trajectories in the front hemisphere, i.e., Stage I (please see Fig. 2 in our paper) remains unchanged, run MVDs only along the up and down orbits in Stage II, and skip Stage III. This way, the proposed pipeline can be generalized to forward-facing scenes. We will add these descriptions along with the experimental results on forward-facing scenes to the final version of our manuscript.
> > >
> > > Additionally, the proposed PVO process, which is the core contribution of our approach, can be generalized to various existing MVD-based methods, such as LiftImage3D. As shown in Figure 3 of the LiftImage3D paper, it generates new views in the forward-facing camera distributions, and we have validated in the ablation study (Section 4.4) in our paper that our PVO process has the generalizability to LiftImage3D and significantly improves its reconstruction quality. This further verifies the ability of our approach on generalizing to forward-facing scenes.
> > >
> > > We would like to thank the reviewer again, and appreciate any follow-up comments and discussions from the reviewer.

---

### Official Review · Reviewer_ExWa · 2025-07-02

**Clarity:** 3
**Significance:** 2
**Originality:** 2
**Rating:** 4
**Confidence:** 3

**Summary:**

This paper addresses the challenge of omnidirectional 3D scene reconstruction from a single image, identifying common issues in existing diffusion-based methods such as geometric distortion and inconsistencies. To overcome these, the authors propose Omni3D, which leverages pose estimation priors and a Pose-View Optimization (PVO) module to iteratively refine novel views and their estimated camera poses. Experimental results demonstrate significant improvement over existing state-of-the-art methods.

**Questions:**

1. This submission is very similar to LiftImage3D from motivation to method. Please clarify the difference between the two, which is very important to highlight the contribution of this submission.

2. Are the experimental setup and comparison methods consistent? Why are the results reported in this article so much lower than those reported in the original paper? It is recommended to unify the evaluation, as this is crucial to enhancing the credibility of the submission. If the uncertainty is due to random selection, then repeated verification is necessary.

3. Include more detailed discussions about computational costs and potential optimizations or approximations to reduce inference time.

My main concern is the contribution of this paper and the reliability of the experimental data. Currently, the contribution of this paper is incremental compared to previous methods, and the experimental results do not appear convincing. I am happy to see the author's clarification and response.

**Ethical Concerns:**

["NO or VERY MINOR ethics concerns only"]

**Final Justification:**

My main concerns were addressed well. The core contribution of this submission is the proposed Pose-View Optimization (PVO) method, which enables 360-degree reconstruction of the scene. This is both meaningful and impressive, so I've increased my initial rating.

**Limitations:**

1. Omni3D may require substantial computational resources, potentially limiting deployment in real-time or resource-constrained scenarios.

2. The method's iterative refinement strategy may accumulate errors, especially for extreme viewpoints or highly complex scene geometries.

**Quality:**

2

**Strengths And Weaknesses:**

Strengths:

1. The task of single-image 3D reconstruction is significant for future AI-powered immersive media. The paper clearly identifies critical issues in current methods and proposes targeted solutions.

2. The iterative Pose-View Optimization (PVO) approach, which jointly optimizes pose estimation and generated views by minimizing 3D reprojection errors, is well-motivated.

3. The video in the supplementary material shows good performance.

Weaknesses:

1. This submission is very similar to LiftImage3D from motivation to methodology. The differences between Omni3D and LiftImage3D must be explicitly clarified to effectively highlight the original contributions of this paper. Currently, the incremental nature of contributions diminishes the novelty of the submission.

2. The reported experimental results are notably lower than those presented in the original papers being compared. This inconsistency raises concerns regarding experimental reliability and evaluation fairness. It is crucial to unify experimental setups and perform repeated evaluations to ensure the credibility of the results.

3. Although parallelization strategies are mentioned, the computational complexity and inference time still appear substantial. The paper lacks detailed discussions on potential optimizations or approximations that could effectively reduce computational costs.

---

> ### Author Rebuttal · Authors · 2025-07-28
>
> Thanks to the reviewer for the positive comments on the motivation of the proposed PVO method and the performance of the videos in the supplementary material. We would like to also thank the reviewer very much for the valuable questions, which are incredibly helpful in improving the quality of our manuscript. We provide point-by-point responses to the reviewer’s comments as follows:
> ### **Novelty and contributions vs. LiftImage3D**
> LiftImage3D proposes a distortion-aware 3DGS method to reduce the distortion of 3DGS during the Gaussian optimization process with the generated multi-view images which suffer from content and geometric inconsistencies. Although the distortion-aware 3DGS mitigates the inconsistencies among multi-view images to a certain degree, it may be an indirect way with incremental effectiveness and low time efficiency. On the contrary, in this paper, we proposed a novel Pose-View Optimization (PVO) method to directly and jointly correct both the estimated poses and the generated contents by an iterative self-supervised learning. This way, the geometric inconsistencies and the error of pose estimations can be directly and effectively refined before the optimization of 3DGS. The experiments validated that our approach is a more effective way with better quality performance and also achieves higher time efficiency (when parallelism applies) than LiftImage3D. More importantly, thanks to the propose PVO method, we are able to achieve the challenging task of *omnidirectional* 3D reconstruction, instead of only reconstructing 3DGS in narrow angles in LiftImage3D.
> ### **Experimental setup**
> Thanks a lot for the valuable comments on the experimental setup. The reason for the lower reported numbers in our manuscript than those reported in the original papers is that we evaluate the quality of rendered images in the entire **omnidirectional** space, instead of only the surrounding views of the input (please see line 11 of Sec. 4.1 in the LiftImage3D paper). We will release the codes publicly with our evaluation protocol, which ensures the reproducibility of our approach, and furthermore may contribute to benchmarking the future works in this direction.
>
> We agree with the reviewer that it is beneficial to conduct additional experiments with the settings aligned with the original paper of compared methods. Since the SOTA method LiftImage3D does not provide an evaluation protocol in their open-sourced codes, we are not able to exactly know their detailed experimental setup, such as which groundtruth views are selected in each sample. Hence, we tried our best to set up the settings where LiftImage3D performs similarly to the original numbers reported in their paper, and then evaluate our proposed Omni3D approach. The results are shown in Table 1 in this rebuttal below.  It can be seen from Table 1 that our Omni3D approach still achieves superior performance under this setting. We will add the results of this experiment to the revised manuscript.
>
> * Table 1: Evaluation on the surrounding views of input (align with LiftImage3D)
>
>     | Methods | Tanks and Temples | Mip-NeRF 360 | DL3DV |
>     |:--------|:-----------------|:------------|:-----|
>     |         | PSNR $\uparrow\ \ \ \ $   SSIM $\uparrow\ \ \ $   LPIPS $\downarrow\ \ \ $ | PSNR $\uparrow\ \ \ \ $  SSIM $\uparrow\ \ \ $  LPIPS     $\downarrow\ \ \ $ | PSNR $\uparrow\ \ \ \ $  SSIM $\uparrow\ \ \ $  LPIPS $\downarrow$ |
>     | ZeroNVS [33]|         13.98 $\ \ \ \ \ $	0.4856 $\ \ \ \ $ 0.6225 |	14.25 $\ \ \ \ \ $	0.2669 $\ \ \ \ $	0.8657|	13.73 $\ \ \ \ \ $	0.5380 $\ \ \ \ $	0.6476
>     | ViewCrafter [51]| 15.79 $\ \ \ \ \ $	0.5054 $\ \ \ \ $	0.5625	 | 15.99 $\ \ \ \ \ $	0.2717 $\ \ \ \ $	0.7012	 | 17.85 $\ \ \ \ \ $	0.6459	$\ \ \ \ $ 0.3843
>     | LiftImage3D [3]|  16.14 $\ \ \ \ \ $	0.5232 $\ \ \ \ $	0.5346 | 	16.09 $\ \ \ \ \ $	0.2776 $\ \ \ \ $ 0.5553 |	20.77 $\ \ \ \ \ $	0.6834 $\ \ \ \ $	0.4557
>     | **Our Omni3D** | **18.32** $\ \ \ \ \ $	**0.5919** $\ \ \ \ $ **0.4689** |	**16.96** $\ \ \ \ \ $	**0.2880** $\ \ \ \ $	**0.4807** | 	**22.02** $\ \ \ \ \ $	**0.7649** $\ \ \ \ $ **0.3011**
>     | | | | | |
>
> ### **Computational costs and potential optimizations**
> We agree with the reviewer that although we employed significant parallelism in computation, the inference time is still substantial. This is mainly because the diffusion-based generation model (i.e., MVD) and Gaussian optimization are time-consuming. Other image-to-3D methods, e.g., ViewCrafter and LiftImage3D, also suffer from the same problem. As Table 3 in our manuscript shows, the proposed PVO method consumes only ~30% of the total computational time and the proposed Omni3D achieves faster speed than LiftImage3D.
> Based on above analysis, a potential way to effectively reduce time consumption is training a (diffusion-based) model to generate 3D Gaussian splats straight from a single image input, bypassing the 2D (multi-view images) intermediate steps. This way, it saves the computational time for generating hundreds of multi-view images and also eliminates the optimization process for 3DGS. As such, the computational complexity is expected to be significantly reduced. This can be seen as interesting and valuable future work.
> ### **Error accumulation in iterative refinement**
> We thank the reviewer very much for pointing out the possible risk of error accumulation. We analyze this issue from two aspects as follows:
> * **Error in the iterative PVO of each pair.** In the iterative PVO process of each pair, the generated views and estimated poses and intrinsics are jointly optimized by minimizing the reprojection error until convergence. This process should *not* suffer from error accumulation, since the error (loss) continuously reduces until nearly unchanged throughout the optimization. In Section C of the *Supplementary Material*, we have conducted the ablation study regarding the quality performance and the iterations of pose updates in PVO. We also show the results as follows. It can be seen from Table 2 in this rebuttal that the reconstruction quality continuously increasing along with the process of iterative PVO (the 0 to 4 iterations of pose updates). This confirms the continuous reduction of error without accumulation. Besides, the performance only negligibly increases at the 4-th iteration, hence it verifies the reasonability for setting the iterations of pose updates to 3 in our PVO method.
>
>     \
>     Table 2: Ablation on iterations of pose updates in PVO, in addition to the initial pose estimation
>     | Iterations | PSNR $\uparrow$ | SSIM $\uparrow$ | LPIPS $\downarrow$ |
>     |---|---|---|---|
>     | 0 (w/o PVO) | 15.56 | 0.5198 | 0.5346 |
>     | 1 | 15.62 | 0.5207 | 0.5325 |
>     | 2 | 15.91 | 0.5254 | 0.5296 |
>     | 3 | 16.30 | 0.5308 | 0.5166 |
>     | 4 | 16.33 | 0.5311 | 0.5162 |
>     | | | | | |
>
> * **Error in the progressive paring scheme.** Recall that in Section 3.3.1, we proposed a progressive pairing scheme with the window size of $N$, i.e., in each generation orbit, the first $N$ generated views undergo PVO with the reference of $\boldsymbol{x}_0$, and then the next $N$ views takes the optimized $N$-th view $\boldsymbol{\hat{x}}_N$ as reference. This progressive scheme continues until all generated views within the orbit have been processed and refined.
>
>     This scheme is a trade-off between the risk of error accumulation and the robustness of PVO. That is, the large size of $N$ (consistently use the initial view as reference for a large range of other views) would lead to progressively larger viewpoint disparities and therefore challenge the robustness of PVO, and on the contrary, using immediately preceding optimized view (very small $N$) may introduce error accumulation. We have conducted the ablation study on this trade-off balance in Section C of the *Supplementary Material*, and we show the results in Table 3 in this rebuttal. It can be seen from Table 3 that utilizing views with large disparities as references (e.g., $N=24$ or $48$) can significantly impair the robustness of pose estimation and diminish the efficacy of PVO. Conversely, employing each immediately preceding optimized view, $\boldsymbol{\hat{x}}_{i-1}$ as the reference for the current view $\boldsymbol{x}_i$ ($N=1$), may inadvertently introduce error accumulation and propagation along the generation path of the orbits. Therefore, setting $N=12$ in our approach results in the best performance, indicating a reasonable balance of the trade-off issue.
>
>     \
>     Table 3: Ablation on progressive pairing
>     | | PSNR $\uparrow$ | SSIM $\uparrow$ | LPIPS $\downarrow$ |
>     |---|---|---|---|
>     | w/o PVO | 15.56 | 0.5198 | 0.5346 |
>     | $N=1$ | 16.24 | 0.5305 | 0.5170 |
>     | $N=12$ | **16.30** | **0.5308** | **0.5166** |
>     | $N=24$ | 16.19 | 0.5281 | 0.5179 |
>     | $N=48$ | 15.98 | 0.5206 | 0.5254 |
>     | | | | | |

---

> > ### Comment · Reviewer_ExWa · 2025-08-05
> > **About Novelty and contributions vs. LiftImage3D**
> >
> > The authors' clarification appears to somewhat sidestep the core issue. The most significant similarity between this submission and LiftImage3D lies in the order of video frame generation. This strategy seems to be a natural and straightforward extension of LiftImage3D, which makes the novelty of the current work appear relatively limited. Furthermore, the authors’ claim that LiftImage3D is only suitable for reconstructing narrow-angle views is unconvincing. Given the adopted frame generation strategy, achieving panoramic views by iteratively updating the reference frame seems feasible—and in fact, this appears to be the starting point of the proposed method. From this perspective, the contribution of the submission is further diminished. I would appreciate a more thorough clarification from the authors.

---

> > > ### Author Response · Authors · 2025-08-05
> > > **Further clarification on novelty and contributions**
> > >
> > > Thank the reviewer very much for the follow-up comments on our rebuttal. Regarding the reviewer’s concerns, we would like to further clarify our novelty and contributions vs. LiftImage3D.
> > >
> > > Indeed, the proposed Omni3D approach employs video frame generation as 2D intermediaries for 3D reconstruction. However, our approach is not a simple extension of LiftImage3D. The core novelty and contribution of our paper is proposing a Pose-View Optimization (PVO) method which empowers the pipeline of video frame generation + 3DGS with the ability to reconstruct the omnidirectional space of 3D scenes with satisfied quality, due to the effectiveness of the proposed PVO on mitigating the distortions, inconsistencies and error accumulations in generated views.
> > >
> > > In Figure 3 of the LiftImage3D paper, it shows that LiftImage3D only generates views in limited angles from the input image, instead of panoramic views. We would argue that straightforwardly extending LiftImage3D by iteratively updating the reference frame of LiftImage3D and repeating video frame generation is not a feasible way for omnidirectional 3D reconstruction, because LiftImage3D does not have the ability to mitigate errors during the multi-view generation. Therefore, the accumulation of content distortions and geometrical inconsistencies prevents LiftImage3D from further extending the generation process to panorama with satisfied quality, and consequently it is not able to reconstruct 3DGS in omnidirectional space.
> > >
> > > On the contrary, we propose the PVO method, which is performed to each generated orbit before it serves as new references, not only mitigates the content and geometrical errors within each orbits, and furthermore stops error accumulations. As such, the propose PVO method plays an essential role to enable us to extend the multi-view generation to cover the panoramic space, and hence empowers our proposed Omni3D approach to pioneer in effective omnidirectional 3D scene reconstruction from a single image.
> > >
> > > In the revised manuscript, we will make a more thorough clarification on the novelty and contributions of the paper.
> > >
> > > We would like to thank the reviewer again for the comment, and appreciate any follow-up comments and discussions from the reviewer.

---

> ### Comment · Reviewer_ExWa · 2025-08-06
>
> Thanks to the authors for further clarification. It is recommended that the authors provide more rendering results of the “back side” of reference views in the manuscript to support the claim of omnidirectional spatial rendering. Anyway, I will raise my rating to "Borderline accept".

---

> ### Author Response · Authors · 2025-08-06
>
> We would like to sincerely thank the reviewer for the valuable and insightful comments, which help us a lot for improving the quality of our paper. We will follow the reviewer's suggestion to provide more rendering results of the back side of the input view in the experiments section. We also thank the reviewer for the willingness to upgrade the rating.

---

### Official Review · Reviewer_Wu2D · 2025-07-03

**Clarity:** 3
**Significance:** 3
**Originality:** 3
**Rating:** 4
**Confidence:** 4

**Summary:**

The paper proposes a novel method for single-image to 3D reconstruction, outperforming prior work. The method uses a multi-view diffusion model to create many views from an input image. The authors then propose a "Pose-View Optimization module" to refine both the camera poses (estimated via MASt3R) and the generated views. This process starts with nearby views and then expands to farther views. Finally, once this has been done for each region (both nearby and far views), 3DGS is used for 3D reconstruction.

**Questions:**

Overall, I like this paper and am inclined towards acceptance. My main concern is I did not notice much improvement in the qualitative results, but I still feel the ideas from the method are a useful contribution.

**Ethical Concerns:**

["NO or VERY MINOR ethics concerns only"]

**Limitations:**

yes

**Paper Formatting Concerns:**

None.

**Quality:**

3

**Strengths And Weaknesses:**

**Strengths**
- The proposed method seems well designed. In particular, the design of the progressive pairing strategy is well thought out and seems like a valuable contribution for future work to build upon.
- In addition to the point above, I appreciated the ablation on the Pose-View Optimization module, as it helped support the benefit of the module.
- The quantitative results support the efficacy of the proposed method.

**Weaknesses**
- I was a bit underwhelmed by the qualitative results. The paper would benefit from a user study to see if the reported metrics (PSNR/SSIM/LPIPS) align. For me, I did not notice significant improvements in the images from the proposed method. Perhaps identifying areas that are significantly better (with a box) and showing a zoomed in view of that region would be helpful for readers.
- Additional ablations on the progressive pairing strategy and different hyperparameters would be informative.

---

> ### Author Rebuttal · Authors · 2025-07-26
>
> We would like to thank the reviewer very much for the constructive suggestions and insights into the qualitative results and the ablations on progressive pairing and hyperparameters.
> ### **Qualitative results**
> * **User study.** According to the suggestion of the reviewer, we conducted a user study with 10 non-expert users, who are requested to rate the reconstructed 3D scenes with scores from 0 (poorest quality) to 10 (perfect quality). In the user study, we render the images with omnidirectional trajectories from the 3DGS generated by our Omni3D and the compared methods and send them to the users in video format, to reduce the hardware requirements for the users’ personal computers. The average ratings are shown in Table 1. It can be seen from Table 1 that our Omni3D approach has obvious superior perceptual quality performance, which aligns with the numerical results reported in the paper. We will add this experiment to the revised manuscript.
>
>     \
>     Table 1: Results of user study on our Omni3D and compared methods
>
>     | Methods | Tanks and Temples$\ \ \ \ \ $ | Mip-NeRF 360$\ \ \ \ \ $ | DL3DV |
>     |:--------|:-----------------|:------------|:-----|
>     | ZeroNVS [33]|    1.0     |        1.3         |          0.8          |
>     | ViewCrafter [51]|       4.3          |        4.7         |          7.4               |
>     | LiftImage3D [3]|         5.1        |        4.5         |          5.8          |
>     | **Our Omni3D** |        **7.6**         |         **7.9**        |          **8.2**          |
>     | | | | | |
>
>
> * **Visual examples in videos.** Besides, we have uploaded three visual examples in video format in the *Supplementary Material*. It can be obviously seen from the videos that the proposed Omni3D approach achieves 3D scene reconstruction in the omnidirectional space with higher visual quality and better geometry consistency than the compared approaches.
> * **Figure 3 in the paper.** In the revised manuscript, we will follow the reviewer’s suggestion to highlight the regions with obvious differences with zoomed in views to increase readability.
> ### **Ablations on the progressive pairing strategy and hyperparameters**
> We have provided additional ablation studies on the progressive pairing strategy and hyperparameters in Section C in the *Supplementary Material*. We also show the results in the following.
> * **Ablation study on progressive pairing.**
> Recall that in Section 3.3.1, we proposed a progressive pairing scheme with the window size of $N$, i.e., in each generation orbit, the first $N$ generated views undergo PVO with the reference of $\boldsymbol{x}_0$, and then the next $N$ views takes the optimized $N$-th view $\boldsymbol{\hat{x}}_N$ as reference. This progressive scheme continues until all generated views within the orbit have been processed and refined.
>
>     Table 2 shows the ablation study on the selection of $N$ on the Tanks and Temples dataset [13]. It can be seen from Table 2 that $N=12$ results in the best performance. This outcome is likely attributable to the fact that utilizing views with large disparities as references (e.g., $N=24$ or $48$) can significantly impair the robustness of pose estimation and diminish the efficacy of PVO. Conversely, employing each immediately preceding optimized view, $\boldsymbol{\hat{x}}_{i-1}$ as the reference for the current view $\boldsymbol{x}_i$ ($N=1$), may inadvertently introduce error accumulation and propagation along the generation path of the orbits. Besides, setting $N=1$ also considerably limits parallelism.
>
>     \
>     Table 2: Ablation on progressive pairing
>     | | PSNR $\uparrow$ | SSIM $\uparrow$ | LPIPS $\downarrow$ |
>     |---|---|---|---|
>     | w/o PVO | 15.56 | 0.5198 | 0.5346 |
>     | $N=1$ | 16.24 | 0.5305 | 0.5170 |
>     | $N=12$ | **16.30** | **0.5308** | **0.5166** |
>     | $N=24$ | 16.19 | 0.5281 | 0.5179 |
>     | $N=48$ | 15.98 | 0.5206 | 0.5254 |
>     | | | | | |
>
> * **Ablation study on iterations of pose updates in PVO.**
> Recall that in Section 3.3.2, we proposed a pairwise iterative PVO method. In the proposed PVO, given the initially estimated poses and intrinsics, we refine the generated view by minimizing the reprojection error until convergence, and then the poses and intrinsics are updated based on the refined views. This cycle is iteratively conducted until estimated poses converge. Table 3 illustrates the performance on the Tanks and Temples dataset [13] with different iterations of pose updates, in addition to the initial pose estimation. It can be seen that the performance converges at 3 iterations, i.e., optimize the views and then update poses for three time. When the iteration number comes to 4, the performance negligibly increases. This verifies the reasonability for setting the iterations to 3 in our approach.
>
>     \
>     Table 3: Ablation on iterations of pose updates in PVO, in addition to the initial pose estimation
>
>     | Iterations | PSNR $\uparrow$ | SSIM $\uparrow$ | LPIPS $\downarrow$ |
>     |---|---|---|---|
>     | 0 (w/o PVO) | 15.56 | 0.5198 | 0.5346 |
>     | 1 | 15.62 | 0.5207 | 0.5325 |
>     | 2 | 15.91 | 0.5254 | 0.5296 |
>     | 3 | 16.30 | 0.5308 | 0.5166 |
>     | 4 | 16.33 | 0.5311 | 0.5162 |
>     | | | | | |

---

> > ### Comment · Reviewer_Wu2D · 2025-08-06
> >
> > I'd like to thank the authors for addressing my questions and concerns through their rebuttal. Despite some flaws, the majority of my concerns were mitigated. I would like to maintain my borderline accept rating.

---

> > > ### Author Response · Authors · 2025-08-06
> > >
> > > We sincerely thank the reviewer for the valuable and insightful comments, that facilitates us improving the quality of our paper. We are glad that our rebuttal has addressed the majority of the reviewer's concerns.

---

> > > ### Author Response · Authors · 2025-08-08
> > > **Further clarification regarding qualitative results of Figure 3 in our paper**
> > >
> > > Thanks again for the reviewer's follow-up comments.
> > >
> > > We would like to make some further clarifications regarding the qualitative results of Figure 3 in our paper. In Figure 3, our rendered images not only have higher visual quality and less artifacts than compared methods, but also are with higher geometrical consistencies (location, angle of view, etc.) with the groundtruth, which is also an essential factor for qualitative quality of 3D reconstruction.
> > >
> > > For example, in the second row of Figure 3, it can be obviously seen that the proposed method achieves the best geometrical consistency with the groundtruth, e.g., the locations of the table and the flowers, and the angle of view of the image, align better with the groudntruth. We will make further clarifications on the issue in the revised manuscript.
> > >
> > > Besides, we also achieves better visual quality with less artifacts in all examples in Figure 3. As mentioned in the previous rebuttal, we will follow the reviewer’s suggestion to highlight the regions with obvious differences with zoomed in views to increase readability.
> > >
> > > Additionally, we will add the results of the user study in the revised manuscript, which has been shown in the previous rebuttal.

---

### Comment · Area_Chair_xLwX · 2025-08-05

Dear Reviewers,

The discussion period deadline is approaching. Please kindly participate to ensure a fair and smooth review process.

Thank you.

Best,

AC

---

### Note · Authors · 2025-08-12

We would like to thank the AC and the reviewers for the efforts on reviewing our manuscript and the valuable and insightful comments.

The reviewers pointed out that omnidirectional 3D reconstruction from a single image is a new (xEyg) and significant (ExWa) task, and praised our method as well-designed and valuable (Wu2D), and the proposed PVO process as an interesting (xEyg), well-motivated (ExWa) and innovative (yK7z) design. The reviewers also highlighted the comprehensiveness of our experiments (xEyg) which shows promising quantitative results (Wu2D), and the videos in the supplementary show good performance (ExWa). The parallelism (yK7z), generalizability and paper writing (yK7z/Exyg) also receive positive comments from the reviewers.

During the rebuttal, we additionally conducted a **user study**, made **evaluations on forward-facing views**, added **ablations for the lightweight CNN**, reported **computational time on a single GPU**, and made clarifications on novelty and contributions, generalizability, the pipeline, Homography, evaluation protocol, details of MVD, etc, to enhance the quality of our paper.

Finally, regarding the main concerns of the reviewers, we would like to briefly make some further clarifications:
* **Novelty.** The core contribution of our paper is proposing a novel Pose-View Optimization (PVO) method, which synergistically optimizes the generated contents and its estimated poses, to mitigate distortions and geometric inconsistencies, and furthermore, prevents error accumulations. It empowers our proposed Omni3D approach to pioneer in effective omnidirectional 3D scene reconstruction from a single image.
* **Complexity.** Although we consume 46.2% more time than LiftImage3D on a single GPU, the effective area and space that we can reconstruct is **several times** larger. Therefore, our **time efficiency per reconstructible space** is obviously higher than LiftImage3D. Moreover, the proposed method can significantly benefits from parallelism, making us faster than LiftImage3D on 8 x A100 GPUs.
* **Qualitative results.** In Figure 3, not only visual quality but also **geometrical consistencies** are essential for 3D qualitative quality. In the second row of Figure 3, it can be obviously seen that we achieve the best geometrical consistency with the groundtruth, e.g., locations of the table and flowers, the angle of view of the image, etc. Besides, we also achieve better visual quality with less artifacts in all examples.

---

### Decision · Program_Chairs · 2025-09-17

**Decision:**

Accept (poster)

**Comment:**

This paper introduces Omni3D, a framework for reconstructing omnidirectional 3D scenes from a single image. The key contribution is the Pose–View Optimization (PVO) module, which jointly refines diffusion-generated views and their estimated poses to mitigate distortions and inconsistencies before reconstruction with 3D Gaussian splatting. The method is evaluated on multiple datasets and demonstrates clear improvements over strong baselines, supported by quantitative metrics, qualitative comparisons, and a user study.

The reviewers appreciated the significance of the task, the clarity of presentation, and the systematic design of the proposed PVO. The work was also praised for its comprehensive experiments, generalizability to different diffusion backbones, and supplementary visualizations. Concerns were raised regarding the incremental nature of the contributions, as parts of the pipeline resemble LiftImage3D, as well as the complexity and efficiency of the overall system. However, the authors addressed these points in rebuttal with additional experiments (forward-facing views, ablations, runtime clarifications) and a perceptual user study, which strengthened confidence in the results and mitigated most concerns.

Overall, while efficiency remains a limitation, the paper makes a meaningful step forward by demonstrating that omnidirectional 3D reconstruction from a single image is feasible. The PVO framework provides a principled and generalizable solution, and the empirical results substantiate its effectiveness.  The final recommendation by the Area Chairs is to accept the submission.